# PRIVATE DATA STREAM ANALYSIS FOR UNIVERSAL SYMMETRIC NORM ESTIMATION

## ABSTRACT

We study how to release summary statistics on a data stream subject to the constraint of differential privacy. In particular, we focus on releasing the family of *symmetric norms*, which are invariant under sign-flips and coordinate-wise permutations on an input data stream and include $L_p$ norms, $k$-support norms, top-$k$ norms, and the box norm as special cases. Although it may be possible to design and analyze a separate mechanism for each symmetric norm, we propose a general parametrizable framework that differentially privately releases a number of sufficient statistics from which the approximation of all symmetric norms can be simultaneously computed. Our framework partitions the coordinates of the underlying frequency vector into different levels based on their magnitude and releases approximate frequencies for the "heavy" coordinates in important levels and releases approximate level sizes for the "light" coordinates in important levels. Surprisingly, our mechanism allows for the release of an *arbitrary* number of symmetric norm approximations without any overhead or additional loss in privacy. Moreover, our mechanism permits $(1 + \alpha)$-approximation to each of the symmetric norms and can be implemented using sublinear space in the streaming model for many regimes of the accuracy and privacy parameters.

## 1 INTRODUCTION

The family of $L_p$ norms represent important statistics on an underlying dataset, where the $L_p$ norm of an $n$-dimensional vector freqeuncy $x$ is defined as the number of nonzero coordinates of $x$ for $p = 0$ and $L_p(x) = (x_1^p + \ldots + x_n^p)^{1/p}$ for $p > 0$. Thus, $L_0$ norm counts the number of distinct elements in the dataset and, e.g., is used to detect denial of service or port scan attacks in network monitoring (Akella et al., 2003; Estan et al., 2003), to understand the magnitude of quantities such as search engine queries or internet graph connectivity in data mining (Palmer et al., 2001), to manage workload in database design (Finkelstein et al., 1988), and to select a minimum-cost query plan in query optimization (Selinger et al., 1979). The $L_1$ norm computes the total number of elements in the dataset and, e.g., is used for data mining (Cormode et al., 2005) and hypothesis testing (Indyk & McGregor, 2008), while the $L_2$ norm, e.g., is used for training random forests in machine learning (Breiman, 2001), computing the Gini index in statistics (Lorenz, 1905; Gini, 1912), and network anomaly detection in traffic monitoring (Krishnamurthy et al., 2003; Thorup & Zhang, 2004). Consequently, $L_p$ estimation has been extensively studied in the data stream model (Alon et al., 1999; Indyk & Woodruff, 2005; Indyk, 2006; Li, 2008; Kane et al., 2011; Andoni, 2017; Braverman et al., 2018b; Ganguly & Woodruff, 2018; Woodruff & Zhou, 2020; 2021). The simplest streaming model is perhaps the insertion-only model, in which a sequence of $m$ updates increments coordinates of an $n$-dimensional frequency vector $x$ and the goal is to compute or approximate some statistic of $x$ in space that is sublinear in both $m$ and $n$.

In many cases, the underlying dataset contains sensitive information that should not be leaked. Hence, an active line of work has focused on estimating $L_p$ norms for various values of $p$, while preserving differential privacy (Mir et al., 2011; Blocki et al., 2012; Smith et al., 2020; Bu et al., 2021; Wang et al., 2021).

**Definition 1.1** (Differential privacy). *(Dwork et al., 2006) Given $\varepsilon > 0$ and $\delta \in (0, 1)$, a randomized algorithm $\mathcal{A} : \mathfrak{U}^* \to \mathcal{Y}$ is $(\varepsilon, \delta)$-differentially private if, for every neighboring streams $\mathfrak{S}$ and $\mathfrak{S}'$*

*and for all $E \subseteq \mathcal{Y}$,*

$$\mathbf{Pr}\left[\mathcal{A}(\mathfrak{S}) \in E\right] \leq e^{\varepsilon} \cdot \mathbf{Pr}\left[\mathcal{A}(\mathfrak{S}') \in E\right] + \delta,$$

*where streams $\mathfrak{S}$ and $\mathfrak{S}'$ are neighboring if there exists a single update $i \in [m]$ such that $u_i \neq u_i'$, where $u_1, \ldots, u_m$ are the updates of $\mathfrak{S}$ and $u_1', \ldots, u_m'$ are the updates of $\mathfrak{S}'$.*

For example, (Blocki et al., 2012) showed that the Johnson-Lindenstrauss transformation preserves differential privacy (DP), thereby showing one of the main techniques in the streaming model for $L_2$ estimation already guarantees DP. Similarly, (Smith et al., 2020) showed that the Flajolet-Martin sketch, which is one of the main approaches for $L_0$ estimation in the streaming model, also preserves DP. Notably, algorithmic designs for $L_p$ estimation in the streaming model differ greatly and require individual analysis to ensure DP, which can be quite difficult due to the complexity of the various techniques. This is especially pronounced in the work of (Wang et al., 2021), who studied the $p$-stable sketch that estimates the $L_p$ norm for $p \in (0, 2]$ (Indyk, 2006)[1]. (Wang et al., 2021) showed that for $p \in (0, 1]$, the $p$-stable sketch preserves DP, but was unable to show DP for $p \in (1, 2]$, even though the general algorithmic approach remains the same. Thus the natural question is whether differential privacy can be guaranteed for an approach that simultaneously estimates the $L_p$ norm in the streaming model, for all $p$. More generally, the family of $L_p$ norms are all symmetric norms, which are invariant under sign-flips and coordinate-wise permutations on an input data stream. Symmetric norms thus also include other important families of norms such as the $k$-support norms and the top-$k$ norms.

In this paper, we show that not only does there exist a differentially private algorithm for the estimation of symmetric norms in the streaming model, but also that there exists an algorithm that privately releases a set of statistics, from which estimates of all (properly parametrized) symmetric norms can be simultaneously computed. To illustrate the difference, suppose we wanted to release approximations of the $L_p$ norm of the stream for $k$ different values of $p$. To guarantee $(\varepsilon, \delta)$-DP for the set of $k$ statistics, we would need, by advanced composition, to demand $\left(\mathcal{O}\left(\frac{\varepsilon}{\sqrt{k}}\right), \mathcal{O}\left(\frac{\delta}{k}\right)\right)$-DP from $k$ instances of a single differentially private $L_p$-estimation algorithm, corresponding to the $k$ different values of $p$. Due to accuracy-privacy tradeoffs, the quality of the estimation will degrade severely as $k$ increases. By comparison, our algorithm releases a single set $C$ of private statistics. By post-processing, we can then estimate the $L_p$ norms for $k$ different values of $p$ while only requiring $(\varepsilon, \delta)$-DP from $C$. Hence, our algorithm can simultaneously handle any large number of estimations of symmetric norms without compromising the quality of approximation.

**Theorem 1.2.** *There exists a $(\varepsilon, \delta)$-differentially private algorithm that outputs a set $C$, from which the $(1 + \alpha)$-approximation to any norm with* maximum modulus of concentration *at most $M$ can be computed, with probability at least $1 - \delta$. The algorithm uses $M^2 \cdot \text{poly}\left(\frac{1}{\alpha}, \frac{1}{\varepsilon}, \log n, \log \frac{1}{\delta}\right)$ bits of space.*

The maximum modulus of concentration of a norm measures the worst-case ratio of the maximum value to the median value of a norm on the $L_2$-unit sphere for any restriction of the coordinates and can intuitively quantify the complexity of computing a norm. For example, the $L_1$ norm is generally "easy" to compute and has maximum modulus of concentration $\mathcal{O}(\log n)$.

We emphasize that prior to our work, there is no algorithm that can handle private symmetric norm estimation, much less simultaneously for all parametrized symmetric norms. Although there is specific analysis for various norm estimation algorithms, e.g., see the discussion on related work in the supplementary material, these algorithms require a specific predetermined norm for their input. Thus a separate private algorithm must be run for each estimation, which increases the overall space. Moreover, for a large number of queries, the privacy parameter will need to be much smaller due to the composition of privacy, and thus to ensure privacy, the utility of each algorithm is provably poor. Our algorithm sidesteps both the space and accuracy problems and is the first and only work to do so.

**Applications.** We briefly describe a number of specific symmetric norms that are handled by Theorem 1.2 and commonly used across various applications in machine learning. We first note the following parameterization of the previously discussed $L_p$ norms.

---

[1] $L_p$ for $p \in (0, 1)$ does not satisfy the triangle inequality and therefore is not a norm, but is still well-defined/well-motivated and can be computed

**Lemma 1.3.** *(Milman & Schechtman, 2009; Klartag & Vershynin, 2007) For $L_p$ norms, we have that $\mathrm{mmc}(L) = \mathcal{O}(\log n)$ for $p \in [1, 2]$ and $\mathrm{mmc}(L) = \mathcal{O}(n^{1/2 - 1/p})$ for $p > 2$.*

Thus our algorithm immediately introduces a differentially private mechanism for the approximation of $L_p$ norms that unlike previous work, e.g., (Blocki et al., 2012; Sheffet, 2019; Choi et al., 2020; Smith et al., 2020; Bu et al., 2021; Wang et al., 2021), does not need to provide separate analysis for specific values of $p$. Moreover for constant-factor approximation, the space complexity is tight with the *optimal $L_p$-approximation* algorithms that do not consider privacy, up to polylogarithmic factors (Kane et al., 2010; Li & Woodruff, 2013; Ganguly, 2015; Woodruff & Zhou, 2021).

**Definition 1.4** ($Q$-norm and $Q'$-norm). *We call a norm $L$ a $Q$-norm if there exists a symmetric norm $L'$ such that $L(x) = L'(x^2)^{1/2}$ for all $x \in \mathbb{R}^n$. Here, we use $x^2$ to denote the coordinate-wise square power of $x$. We also call a norm $L'$ a $Q'$-norm if its dual norm is a $Q$-norm.*

The family of $Q'$-norms includes the $L_p$ norms for $1 \le p \le 2$, the $k$-support norm, and the box norm (Bhatia, 2013) and thus $Q'$-norms have been proposed to regularize sparse recovery problems in machine learning. For instance, (Argyriou et al., 2012) showed that $Q'$ norms have tighter relaxations than elastic nets and can thus be more effective for sparse prediction. Similarly, (McDonald et al., 2014) used $Q'$ norms to optimize sparse prediction algorithms for multitask clustering.

**Lemma 1.5.** *(Blasiok et al., 2017) $\mathrm{mmc}(L) = \mathcal{O}(\log n)$ for every $Q'$-norm $L$.*

Theorem 1.2 and Lemma 1.5 thus present a differentially private algorithm for $Q'$-norm approximation that uses polylogarithmic space.

**Definition 1.6** (Top-$k$ norm). *The top-$k$ norm for a vector $x \in \mathbb{R}^n$ is the sum of the largest $k$ coordinates of $|x|$, where we use $|x|$ to denote the coordinate-wise absolute value of $x$.*

The top-$k$ norm is frequently used to understand the more general Ky Fan $k$-norm (Wu et al., 2014), which is used to regularize optimization problems in numerical linear algebra. Whereas the Ky Fan $k$ norm is defined as the sum of the $k$ largest singular values of a matrix, the top-$k$ norm is equivalent to the Ky Fan $k$ norm when the input vector $x$ represents the vector of the singular values of the matrix.

**Lemma 1.7.** *(Blasiok et al., 2017) $\mathrm{mmc}(L) = \tilde{\mathcal{O}}\left(\sqrt{\frac{n}{k}}\right)$ for the top-$k$ norm $L$.*

In particular, the top-$k$ norm for a vector of singular values when $k = n$ is equivalent to the Schatten-1 norm of a matrix, which is a common metric for matrix fitting problems such as low-rank approximation (Li & Woodruff, 2020).

## 2 PRELIMINARIES

In this section, we introduce definitions and simple or well-known results from differential privacy, sketching algorithms, and symmetric norms. For notation, we use $[n]$ for an integer $n > 0$ to denote the set $\{1, \ldots, n\}$. We also use the notation $\mathrm{poly}(n)$ to represent a constant degree polynomial in $n$ and we say an event occurs *with high probability* if the event holds with probability $1 - \frac{1}{\mathrm{poly}(n)}$. Similarly, we use $\mathrm{polylog}(n)$ to denote $\mathrm{poly}(\log n)$.

**Sketching algorithms.** Given a frequency vector $x \in \mathbb{R}^n$ on a data stream, the AMS algorithm for $L_2$-estimation first generates a sign vector $\sigma \in \{-1, +1\}^n$ and sets $S_1 = (\langle \sigma, x \rangle)^2$. The AMS algorithm then repeats this process $b = \frac{6}{\alpha^2}$ independent times to obtain dot products $S_1, \ldots, S_b$, sets $Z^2$ to be the arithmetic mean of $S_1, \ldots, S_b$, and reports $Z$.

We define the $L_2$ norm of a vector $x \in \mathbb{R}^n$ by $L_2(x) = \sqrt{x_1^2 + \ldots + x_n^2}$.

**Definition 2.1** ($\nu$-approximate $\eta$ $L_2$-heavy hitters). *Given an accuracy parameter $\nu \in (0, 1)$, a threshold parameter $\eta$, and a frequency vector $x \in \mathbb{R}^n$, compute a set $H$ and a set of approximations $\widehat{x_k}$ for all $k \in H$ such that:*

*(1) If $x_k \ge \eta L_2(x)$ for any $k \in [n]$, then $k \in H$, so that $H$ contains all $\eta$ $L_2$-heavy hitters of $x$.*

*(2) There exists a universal constant $C > 0$ so that if $x_k \le \frac{C\eta}{2} L_2(x)$ for any $k \in [n]$, then $k \notin H$, so that $H$ does not contain any index that is not an $\frac{C\eta}{2}$ $L_2$-heavy hitter of $x$.*

*(3) If $k \in H$ for any $k \in [n]$, then compute $(1 \pm \nu)$-approximation to the frequency $x_k$, i.e., a value $\widehat{x_k}$ such that $(1 - \nu)x_k \leq \widehat{x_k} \leq (1 + \nu)x_k$.*

We introduce and use a private variant PRIVCOUNTSKETCH of the well-known COUNTSKETCH algorithm (Charikar et al., 2004) by adding noise to each coordinate and then using a standard private threshold routine to ensure differential privacy. Specifically, PRIVCOUNTSKETCH first uses the COUNTSKETCH data structure to obtain an estimated frequency for each coordinate. It then adds Laplacian noise with scale parameter $\mathcal{O}\left(\frac{1}{\eta^2 \nu^2}\right)$ to each estimated frequency and then acquires a threshold $T$ from the $L_2$ norm estimation algorithm AMS and releases all coordinates (and estimated frequencies) whose estimated frequencies are at least $\frac{\nu \eta T}{2} + X$, where $X$ is Laplacian noise with scale parameter $\mathcal{O}\left(\frac{1}{\eta^2 \nu^2}\right)$. Then PRIVCOUNTSKETCH gives the following guarantees:

**Lemma 2.2.** *There exists a one-pass streaming algorithm PRIVCOUNTSKETCH that takes an accuracy parameter $\nu \in (0, 1)$ and a threshold parameter $\eta^2$ and outputs a list $H$ that contains all indices $k \in [n]$ of an underlying frequency vector $x$ with $x_k \geq \eta L_2(x)$ and no index $k \in [n]$ with $x_k \leq \eta(1 - \nu) L_2(x)$. For each $k \in H$, PRIVCOUNTSKETCH also reports a estimated frequency $\widehat{x_k}$ such that $(1 - \nu)x_k - \mathcal{O}\left(\frac{\log m}{\eta \nu}\right) \leq \widehat{x_k} \leq (1 + \nu)x_k + \mathcal{O}\left(\frac{\log m}{\eta \nu}\right)$. The algorithm uses $\mathcal{O}\left(\frac{1}{\eta^2 \nu^2} \log^2 n\right)$ bits of space and succeeds with probability $1 - \frac{1}{\text{poly}(m)}$.*

**Symmetric norms.** We now introduce preliminaries for symmetric norms.

**Definition 2.3** (Symmetric norm). *A function $L : \mathbb{R}^n \to \mathbb{R}$ is a symmetric norm if $L$ is a norm and for all $x \in \mathbb{R}^n$ and any vector $y \in \mathbb{R}^n$ that is a permutation of the coordinates of $x$, we have $L(x) = L(y)$. Moreover, we have $L(x) = L(|x|)$, where $|x|$ is the coordinate-wise absolute value of $x$.*

**Definition 2.4** (Modulus of concentration). *Let $x \in \mathbb{R}^n$ be a random variable drawn from the uniform distribution on the $L_2$-unit sphere $S^{n-1}$ and let $b_L$ denote the maximum value of $L(x)$ over $S^{n-1}$. The median of a symmetric norm $L$ is the unique value $M_L$ such that $\mathbf{Pr}\left[L(x) \geq M_L\right] \geq \frac{1}{2}$ and $\mathbf{Pr}\left[L(x) \leq M_L\right] \geq \frac{1}{2}$. Then the ratio $\text{mc}(L) := \frac{b_L}{M_L}$ is the modulus of concentration of the norm $L$.*

Although the modulus of concentration quantifies the "average" behavior of the norm $L$ on $\mathbb{R}^n$, norms with challenging behavior can still be embedded in lower-dimensional subspaces. For instance, the $L_1$ norm satisfies $\text{mc}(L) = \mathcal{O}(1)$, but when $x \in \mathbb{R}^n$ has fewer than $\sqrt{n}$ nonzero coordinates, the norm $\max(L_\infty(x), L_1(x)/\sqrt{n})$ on the unit ball becomes identically $L_\infty(x)$ (Blasiok et al., 2017), which requires $\Omega(\sqrt{n})$ space (Alon et al., 1999) to estimate. Hence, we further quantify the behavior of a norm $L$ by examining its behavior on all lower dimensions.

**Definition 2.5** (Maximum modulus of concentration). *For a norm $L : \mathbb{R}^n \to \mathbb{R}$ and every $k \leq n$, define the norm $L^{(k)} : \mathbb{R}^k \to \mathbb{R}$ by $L^{(k)}((x_1, \ldots, x_k)) := L((x_1, \ldots, x_k, 0, \ldots, 0))$. Then the maximum modulus of concentration of the norm $L$ is $\text{mmc}(L) := \max_{k \leq n} \text{mc}(L^{(k)}) = \max_{k \leq n} \frac{b_{L^{(k)}}}{M_{L^{(k)}}}$.*

**Definition 2.6** (Important Levels). *For $x \in \mathbb{R}^n$ and $\xi > 1$, we define the level $i$ as the set $B_i = \{k \in [n] : \xi^{i-1} \leq |x_k| \leq \xi^i\}$. We define $b_i := |B_i|$ as the size of level $i$. For $\beta \in (0, 1]$, we say level $i$ is $\beta$-important if $b_i > \beta \sum_{j>i} b_j$ and $b_i \xi^{2i} \geq \beta \sum_{j \leq i} b_j \xi^{2j}$.*

Informally, level $i$ is $\beta$-important if (1) its size is at least a $\beta$-fraction of the total sizes of the higher levels and (2) its contribution is roughly a $\beta$-fraction of the total contribution of all the lower levels. We would like to show that to approximate a symmetric norm $L(x)$, it suffices to identify the $\beta$-important levels and their sizes for a fixed base $\xi > 1$.

**Lemma 2.7.** *(Blasiok et al., 2017) Let $s = \mathcal{O}(\log n)$. If a level $i$ is $\beta$-important, then either $\xi^{2i} \geq \frac{\alpha^2 \beta \varepsilon^2}{\log^2 m} F_2(x)$ or there exists $j \in [s]$ such that $b_i \geq \frac{2^j \log^2 m}{\alpha^2 \varepsilon^2}$ and $\xi^{2i} \in \left[\frac{\alpha^2 \beta \varepsilon^2}{\log^2 m} \cdot \frac{F_2(x)}{2^j}, \frac{\alpha^2 \beta \varepsilon^2}{\log^2 m} \cdot \frac{F_2(x)}{2^{j-1}}\right]$.*

Lemma 2.7 implies that if level $i$ is $\beta$-important, then either (1) it will be identified by using PRIVCOUNTSKETCH with threshold $\frac{\alpha^2 \beta}{\log^2 m}$ on the stream or (2) its contribution can be well-

approximated by using PRIVCOUNTSKETCH with threshold $\frac{\alpha^2 \beta \varepsilon^2}{\log^2 m}$ on a substream formed by sampling coordinates of the universe with probability $\frac{1}{2^j}$. We thus split our algorithm and analysis to handle these cases. In particular, we call a frequency level $i$ "high" if $\xi^{2i} \geq \frac{\alpha^2 \beta \varepsilon^2}{\log^2 m} F_2(x)$. We call a frequency level $i$ "medium" if $\xi^{2i} \geq \frac{\alpha^2 \beta' \varepsilon^2}{2^j} F_2(x) > T$ and $b_i \geq \mathcal{O}\left(\frac{2^j \log^2 m}{\alpha^2 \varepsilon^2}\right)$ for a certain $\beta' > 0$ and a threshold $T$. We call a frequency level $i$ "low" if $\xi^{2i} \geq \frac{\alpha^2 \beta' \varepsilon^2}{2^j} F_2(x)$ and $b_i \geq \mathcal{O}\left(\frac{2^j \log^2 m}{\alpha^2 \varepsilon^2}\right)$, but $T \geq \frac{\alpha^2 \beta' \varepsilon^2}{2^j} F_2(x)$.

## 3  ALGORITHMIC INTUITION AND OVERVIEW

In this section, we give a brief technical overview of both our algorithmic intuition and how our approaches differ from previous (non-private) work. We defer the full proofs and additional discussion of related work to the supplementary material.

Our starting point is the $L_p$ estimation algorithm of (Indyk & Woodruff, 2005), which was parametrized by (Blasiok et al., 2017) to handle symmetric norms. For a $(1 + \alpha)$-approximation, the algorithm partitions the $n$ coordinates of the frequency vector $x$ into powers of $\xi$-based on their magnitudes, where $\xi > 1$ is a fixed function of $\alpha$. Each partition forms a level set, but (Indyk & Woodruff, 2005; Blasiok et al., 2017) showed that it suffices to accurately count the size of each *important* level set and zero out to the other level sets, where a level set is considered important if its size is large enough to contribute an $\frac{\alpha^2}{\log m}$ fraction of the symmetric norm.

**Private symmetric norm estimation in the centralized setting.** To preserve $(\varepsilon, \delta)$-differential privacy, one initial approach would be to treat the statistics as a histogram and add Laplacian noise with scale $\mathcal{O}\left(\frac{1}{\varepsilon}\right)$ to the frequency of each element. However, the level sets consisting of elements with frequencies between $[\xi^i, \xi^{i+1})$ for small $i$, say $i = 0$, could be largely perturbed by such Laplacian noise. Fortunately, if $i$ is small, the corresponding level set must contain a large number of elements if it is important, so it seems possible to privately release the size $\Gamma_i$ of the level set. Indeed, we can show that the $L_1$ sensitivity of the vector corresponding to level set sizes is small and so we can add Laplacian noise with scale $\mathcal{O}\left(\frac{1}{\varepsilon}\right)$ to each level set size. Hence if the level set has size $\Gamma_i$ roughly $\Omega\left(\frac{1}{\alpha \varepsilon}\right)$, then the Laplacian noise will affect $\Gamma_i$ by a $(1 + \alpha)$-factor.

Unfortunately, there can be level sets that are both important and small in size. For example, if there is a single element with frequency $m$, then the size of the corresponding level set is just one. Then adding Laplacian noise with scale $\mathcal{O}\left(\frac{1}{\varepsilon}\right)$ will severely affect the size of the level set and thus the estimation of the symmetric norm. On the other hand, for $m > \frac{1}{\alpha \varepsilon}$, the frequency of the coordinate is quite large so again it seems like we can just add Laplacian noise with scale $\mathcal{O}\left(\frac{1}{\varepsilon}\right)$ and output the noisy frequency of the coordinate.

**New approach: classifying and separately handling high, medium, and low frequency levels.** The main takeaway from these challenges is that we should handle different level sets separately. We partition the levels into three groups after defining thresholds $T_1$ and $T_2$, with $T_1 > T_2$. We define the "high frequency levels" as the levels whose coordinates exceed $T_1$ in frequency. The intuition is that because the high frequency levels have such large magnitude, their frequencies can be well-approximated by running an $L_2$-heavy hitters algorithm on the stream $S$.

We define the "medium frequency levels" as the levels whose coordinates are between $T_1$ and $T_2$ in frequency. These coordinates are not large enough to be detected by running an $L_2$-heavy hitters algorithm on the stream $S$. However, the sizes of these level sets must be large if the level set is important. Thus there exists a substream $S_j$ for which a large number of these coordinates are subsampled and their frequencies can be well-approximated by running an $L_2$-heavy hitters algorithm on the substream $S_j$.

Finally, we define the "low frequency levels" as the levels whose coordinates are less than $T_2$ in frequency. These coordinates are small enough that we cannot add Laplacian noise to their frequencies without affecting the level sets they are mapped to. Instead, we show that $L_1$ sensitivity for the level set estimations is particularly small for the low frequency levels. Thus, for these frequency levels, we report the size of the frequency levels rather than the identities of the heavy-hitters. We remark

that if our goal was to just approximate the symmetric norms without preserving differential privacy, then it would suffice to just consider the high and medium frequency levels, since the low frequency levels are particularly problematic when Laplacian noise is added to the frequency vector. We also remark that we only use the thresholds $T_1$ and $T_2$ for the purposes of describing our algorithm – in the actual implementation of the algorithm, the thresholds $T_1$ and $T_2$ will be implicitly defined by each of the substreams. We summarize our new approach in Figure 1.

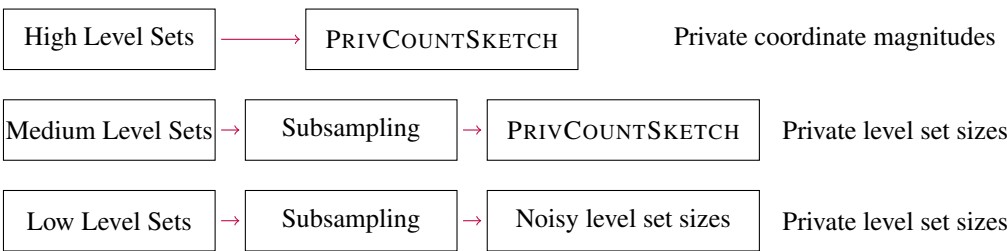

Fig. 1: Illustration of separate handling of the high, medium, and low level sets.

**Private symmetric norm estimation in the streaming model.** Although the previously discussed intuition builds towards a working algorithm, the main caveat is that so far, we have mainly discussed the centralized model, where space is not restricted and so each coordinate and thus each level set size can be counted exactly. In the streaming model, we cannot explicitly track the frequency vector, or even the frequencies of a constant fraction of coordinates. Instead, to estimate the sizes of each level set, (Indyk & Woodruff, 2005; Blasiok et al., 2017) take the stream $S$ and form $s = \mathcal{O}\left(\log n\right)$ substreams $S_1, \ldots, S_s$, where the $j$-th substream is created by sampling the universe of size $n$ at a rate of $\frac{1}{2^{j-1}}$. Then $S_j$ will only consist of the stream updates to the particular coordinates of $x$ that are sampled. Thus in expectation, the frequency vector induced by $S_j$ will have sparsity $\frac{\|x\|_0}{2^{j-1}}$. Similarly, if a level set $i$ has size $\Gamma_i$, then $\frac{\Gamma_i}{2^{j-1}}$ of its members will be sampled in $S_j$ in expectation. It can then be shown through a variance argument that if level set $i$ is important, then there exists an explicit substream $j$ from which $\Gamma_i$ can be well-approximated using the $L_2$-heavy hitter algorithm COUNTSKETCH and as a result, the symmetric norm of $x$ can be well-approximated. The main point of the subsampling approach is that if there exists a level set with large size consisting of small coordinates, then the coordinates will not be detected by the COUNTSKETCH on $S$, but because $S_j$ has significantly smaller $L_2$ norm, then the coordinates will be detected by COUNTSKETCH on $S_j$.

However, adapting the subsampling and heavy-hitter approach introduces additional challenges for privacy. For instance, we can analyze the $L_2$-heavy hitter algorithm COUNTSKETCH and show that although the $L_1$ sensitivity of the estimated frequency for a single coordinate is small, the $L_1$ sensitivity of the estimated frequency for all the coordinates is large. Instead, we use the view that COUNTSKETCH is a composition function that first only estimates frequencies for the top $\text{poly}\left(\frac{1}{\alpha}, \frac{1}{\varepsilon}, \log n\right)$ and then outputs only those estimates that are above a certain threshold. Similarly, the Laplacian noise added to privately use COUNTSKETCH can alter the sizes of a significant number of level sets for small coordinates. Thus for the small coordinates (corresponding to the substreams $S_j$ with large $j$), we invoke COUNTSKETCH with much higher accuracy, so that with high probability, it will return *exactly* the frequencies for the small coordinates. For example, note that if the frequency $f_k$ of a coordinate $k \in [n]$ is at most $\frac{1}{2\alpha^2\varepsilon}$, then any $(1+\alpha^2\varepsilon)$-approximation to $f_k$ can be rounded to exactly recover $f_k$. This decreases the $L_1$ sensitivity of the vector of estimated level set sizes, therefore allowing us to add Laplacian noise without greatly affecting the quality of approximation.

## 4 PRIVATE SYMMETRIC NORM ESTIMATION ALGORITHM

In this section, we give our algorithm that releases a set of private statistics from which an arbitrary number of symmetric norms can be well-approximated. In particular, recall that it suffices to approximate the sizes of the important levels and identity the non-important levels, so that their contributions can be set to zero.

## 4.1 Recovery of High Frequency Levels

As a warm-up, we describe our algorithm for recovering the high frequency levels, whose coordinates have sufficiently large magnitude and thus their frequencies can be well-approximated by running an $L_2$-heavy hitters algorithm on the stream $S$. Moreover, with high probability, adding Laplacian noise will not affect the level sets because the frequencies are so large. Thus it simply suffices to return the noisy estimated frequencies of each of the elements in the high frequency levels. This algorithm is the simplest of our cases and we give the algorithm in full in Algorithm 1.

---

**Algorithm 1** Algorithm to privately estimate the high levels

---

**Input:** Privacy parameter $\varepsilon > 0$, accuracy parameter $\alpha \in (0, 1)$
**Output:** Private estimation of the frequencies of the coordinates of the high frequency levels
1: $\beta \leftarrow \mathcal{O}\left(\frac{\alpha^5}{\mathrm{mmc}(L)^2 \log^5 m}\right)$, $\beta' \leftarrow \mathcal{O}\left(\frac{\alpha^2 \beta \varepsilon^2}{\log^2 m}\right)$
2: Run PRIVCOUNTSKETCH on the stream $S$ with threshold $\alpha^2 \beta'$ and failure probability $\frac{1}{\mathrm{poly}(m)}$
3: **for** each heavy-hitter $k \in [n]$ reported by PRIVCOUNTSKETCH **do**
4:      Let $\widetilde{f_k}$ be the frequency estimated by PRIVCOUNTSKETCH
5:      $\widehat{x_k} \leftarrow \widetilde{x_k} + \mathsf{Lap}\left(\frac{8}{\beta' \varepsilon}\right)$
6:      **return** $\widehat{x_k}$

---

We first show that coordinates in high frequency levels are identified and their frequencies are accurately estimated and similarly that if a coordinate does not have high frequency, it will not be output by Algorithm 1.

**Lemma 4.1.** *Suppose $x_k^2 \geq \frac{\alpha^2 \beta \varepsilon^2}{\log^2 m} F_2(x)$ and $m = \frac{\Omega(\log^5 m)}{\alpha^5 \beta^2 \varepsilon^5}$. Then with high probability, Algorithm 1 outputs $\widehat{x_k}$ such that $(1 - \alpha^2)x_k \leq \widehat{x_k} \leq x_k$. On the other hand if $x_k^2 < \frac{\alpha^2 \beta \varepsilon^2}{2 \log^2 m} F_2(x)$, then with high probability, Algorithm 1 outputs $\widehat{x_k}$ such that $\widehat{x_k} < \frac{3\alpha^2 \beta \varepsilon^2}{4 \log^2 m} F_2(x)$.*

We now justify the privacy and space complexity of Algorithm 1.

**Lemma 4.2.** *Algorithm 1 is $\left(\frac{\varepsilon}{4}, \frac{\delta}{4}\right)$-differentially private for $\delta = \frac{1}{\mathrm{poly}(m)}$ and uses space $\mathrm{mmc}(L)^2 \cdot \mathrm{poly}\left(\frac{1}{\alpha}, \frac{1}{\varepsilon}, \log m\right)$.*

## 4.2 Recovery of Medium Frequency Levels

In this section, we describe our algorithm for recovering the medium frequency levels, whose coordinates do not have sufficiently large magnitude to be detected by running an $L_2$-heavy hitters algorithm on the stream $S$, but have sufficiently large size, so that there exists some $j \in [s]$ across the $s$ subsampling levels such that the coordinates can be detected by running an $L_2$-heavy hitters algorithm on the stream $S_j$. On the other hand, their magnitudes are sufficiently large so that with high probability, adding Laplacian noise will not affect the level sets. We give the algorithm in full in Algorithm 2.

We first upper bound the second frequency moment (and hence the $L_2$ norm) of each substream. This is necessary because we want to detect the coordinates of the medium frequency levels as $L_2$-heavy hitters for each substream, but if the substream has overwhelmingly large $L_2$ norm, then we will not be able to find coordinates of the medium frequency levels. However, it may not be true that $F_2(S_j)$ is significantly smaller than $F_2(S)$ with high probability. For example, if there were a single large element, then the probability it is sampled at level $s$ is $\frac{1}{2^s}$, which is roughly $\frac{1}{n} > \frac{1}{\mathrm{poly}(m)}$. Instead, we note that PRIVCOUNTSKETCH benefits from the stronger *tail guarantee*, which states that not only does PRIVCOUNTSKETCH with threshold $\eta < 1$ detect the elements $k$ such that $(f_k)^2 \geq \eta F_2(S)$, but it also detects the elements $k$ such that $(f_k)^2 \geq \eta F_2(S_{\mathrm{tail}(1/\eta)})$, where $S_{\mathrm{tail}(1/\eta)}$ is the frequency vector $f$ induced by $S$, with the largest $\frac{1}{\eta}$ entries instead set to zero (Braverman et al., 2017; 2018a).

**Lemma 4.3.** *With high probability, $F_2((S_j)_{1/(\alpha^2 \beta' \varepsilon^2)}) \leq \frac{200 \log m}{2^j} F_2(x)$ for all $j \in [s]$.*

---

**Algorithm 2** Algorithm to privately estimate the medium levels

---

**Input:** Privacy parameter $\varepsilon > 0$, accuracy parameter $\alpha \in (0,1)$
**Output:** Private estimations of the sizes of the medium frequency levels

1: $\beta \leftarrow \mathcal{O}\left(\frac{\alpha^5}{\mathrm{mmc}(L)^2 \log^5 m}\right)$, $\beta' \leftarrow \mathcal{O}\left(\frac{\alpha^3 \beta \varepsilon^2}{\log^2 m}\right)$, $\xi \leftarrow (1 + \mathcal{O}(\varepsilon))$

2: $\gamma \leftarrow (1/2, 1)$ uniformly at random, $\ell \leftarrow \lceil \log_\xi(2m) \rceil$, $s \leftarrow \mathcal{O}(\log n)$

3: **for** $j \in [s]$ with $2^j > \frac{\log n}{\beta' \alpha \varepsilon}$ **do**

4:     Form stream $S_j$ by sampling elements of $[n]$ with probability $\frac{1}{2^j}$

5:     Run PRIVCOUNTSKETCH$_j$ on stream $S_j$ with threshold $\alpha^2 \beta' \varepsilon^2$ and failure probability $\frac{1}{\mathrm{poly}(m)}$

6:     **for** each heavy-hitter $k \in [n]$ reported by PRIVCOUNTSKETCH$_j$ **do**

7:         Let $\widehat{x_k}$ be the frequency estimated by PRIVCOUNTSKETCH$_j$

8:         **if** $\widehat{x_k} > \frac{\log n}{\beta' \alpha \varepsilon}$ **then**

9:             $\widetilde{x_k} \leftarrow \widehat{x_k} + \mathsf{Lap}\left(\frac{8}{\beta' \varepsilon}\right)$

10:     **for** $i \in [\ell]$ with $\frac{m^2}{2^{j+1}} > \gamma \xi^{2i} \geq 2^j > \mathcal{O}\left(\frac{\log n}{\beta' \alpha^2 \varepsilon}\right)$ **do**

11:         Let $\widehat{b_i}$ be the number of indices $k \in [n]$ such that $\gamma \xi^{2i} \leq \widetilde{x_k} < \gamma \xi^{2i+2}$

12:         $\widehat{b_i} \leftarrow \frac{2^j}{(1 + \mathcal{O}(\alpha))} \widehat{b_i}$

13:         **return** $\widehat{b_i}$

---

We now show that conditioned on the event that the $L_2$ norm of the subsampled streams are not too large, then we can well-approximate the frequency of any coordinate of the medium frequency levels, provided that they are sampled in the substream.

**Lemma 4.4.** *Suppose $i$ is a $\beta$-important level and $k \in [n]$ is in level $i$, so that $x_k \in [\xi^i, \xi^{i+1})$. If $F_2((S_j)_{1/(\alpha^2 \beta' \varepsilon^2)}) \leq \frac{200 \log m}{2^j} F_2(x)$ for all $j \in [s]$, then $k$ is sampled in stream $S_j$ with $2^j > \frac{\log n}{\beta' \alpha \varepsilon}$, then with high probability, Algorithm 2 outputs $\widehat{x_k}$ such that $(1 - \alpha^2) x_k \leq \widehat{x_k} \leq x_k$.*

Unfortunately, Lemma 4.4 only provides guarantees for the coordinates of the medium frequency levels that are sampled. Thus, we still need to use Lemma 4.4 to show that a good estimator to the sizes of the medium frequency levels can be obtained from the estimates of the coordinates of the medium frequency levels that are sampled. In particular, we show that rescaling the empirical sizes of the medium frequency levels forms a good estimator to the actual sizes of the medium frequency levels.

**Lemma 4.5.** *Consider a $\beta$-important level $i$ with $\xi^{2i} \in \left[\frac{\beta \alpha^2 \varepsilon^2}{\log^2 m} \cdot \frac{F_2(x)}{2^j}, \frac{\beta \alpha^2 \varepsilon^2}{\log^2 m} \cdot \frac{F_2(x)}{2^{j-1}}\right]$ for some integer $j > 0$ and $\xi^i > \frac{\log n}{\beta' \alpha \varepsilon}$. If $F_2((S_j)_{1/(\alpha^2 \beta' \varepsilon^2)}) \leq \frac{200 \log m}{2^j} F_2(x)$ for all $j \in [s]$, then $k$ is sampled in stream $S_j$ with $2^j >$, then with high probability, Algorithm 2 outputs $\widehat{b_i}$ such that $(1 - \mathcal{O}(\alpha)) b_i \leq \widehat{b_i} \leq b_i$, where $b_i$ is the size of level $i$.*

We now analyze the priavcy and the space complexity of Algorithm 2

**Lemma 4.6.** *Algorithm 2 is $\left(\frac{\varepsilon}{4}, \frac{\delta}{4}\right)$-differentially private for $\delta = \frac{1}{\mathrm{poly}(m)}$ and uses space $\mathrm{mmc}(L)^2 \cdot \mathrm{poly}\left(\frac{1}{\alpha}, \frac{1}{\varepsilon}, \log m\right)$.*

## 4.3 RECOVERY OF LOW FREQUENCY LEVELS

In this section, we describe our algorithm for recovering the low frequency levels, whose coordinates have magnitude small enough that we cannot add Laplacian noise to their frequencies without affecting the corresponding level set sizes. We instead report the sizes of the level sets for the low frequency levels rather than the identities and approximate frequencies of the heavy-hitters. Thus we must add Laplacian noise to the sizes of the level sets; we show that $L_1$ sensitivity for the level set estimations is particularly small for the low frequency levels and thus the Laplacian noise does not greatly affect the estimates of the level set sizes. We note that this approach does not work for the high frequency levels because the high frequency levels may have small level set sizes, so that

adding Laplacian noise to the sizes can significantly affect the resulting estimates of the level set sizes. Similarly, it is more challenging to argue the low $L_1$ sensitivity for the level set estimations for the medium frequency levels. Hence, both the algorithm and analysis are especially well-catered to the low frequency levels. We give the algorithm in full in Algorithm 3.

---

**Algorithm 3** Algorithm to privately estimate the low levels

---

**Input:** Privacy parameter $\varepsilon > 0$, accuracy parameter $\alpha \in (0, 1)$
**Output:** Private estimations of the sizes of the low frequency levels
1: $\beta \leftarrow \mathcal{O}\left(\frac{\alpha^5}{\text{mmc}(L)^2 \log^5 m}\right), \beta' \leftarrow \mathcal{O}\left(\frac{\alpha^2 \beta \varepsilon}{\log n}\right), \xi \leftarrow (1 + \mathcal{O}(\varepsilon))$
2: $\gamma \leftarrow (1/2, 1)$ uniformly at random, $\ell \leftarrow \lceil \log_\xi(2m) \rceil, s \leftarrow \mathcal{O}(\log n)$
3: **for** $j \in [s]$ with $2^j \leq \frac{\log n}{\beta' \alpha \varepsilon}$ **do**
4:      Form stream $S_j$ by sampling elements of $[n]$ with probability $\frac{1}{2^j}$
5:      Run PRIVCOUNTSKETCH$_j$ on stream $S_j$ with threshold $\beta'' := \mathcal{O}\left(\frac{\beta' \alpha^2 \varepsilon^3}{\log^2 n}\right)$
6:      **for** each heavy-hitter $k \in [n]$ reported by PRIVCOUNTSKETCH$_j$ **do**
7:          Let $\widehat{x_k}$ be the frequency estimated by PRIVCOUNTSKETCH$_j$
8:      **for** $i \in [\ell]$ with $\mathcal{O}\left(\frac{\log n}{\beta' \alpha^2 \varepsilon}\right) \geq 2^{j+1} > \gamma \xi^{2i} \geq 2^j$ **do**
9:          Let $\widetilde{b}_i$ be the number of indices $k \in [n]$ such that $\gamma \xi^{2i} \leq \widehat{x_k} < \gamma \xi^{2i+2}$
10:          $\widehat{b}_i \leftarrow \frac{2^j}{(1 + \mathcal{O}(\alpha))}\left(\widetilde{b}_i + \mathsf{Lap}\left(\frac{8}{\varepsilon}\right)\right)$
11:          **return** $\widehat{b}_i$

---

We first show that the estimates of the level set sizes for the low frequency levels are accurate.

**Lemma 4.7.** *Consider a $\beta$-important level $i$ with $\xi^{2i} \in \left[\frac{\beta \alpha^2 \varepsilon^2}{\log^2 m} \cdot \frac{F_2(x)}{2^j}, \frac{\beta \alpha^2 \varepsilon^2}{\log^2 m} \cdot \frac{F_2(x)}{2^{j-1}}\right]$ for some integer $j > 0$ and $\xi^i \leq \frac{\log n}{\beta' \alpha \varepsilon}$. If $F_2((S_j)_{1/(\alpha^2 \beta' \varepsilon^2)}) \leq \frac{200 \log m}{2^j} F_2(x)$ for all $j \in [s]$, then $k$ is sampled in stream $S_j$ with $2^j >$, then with high probability, Algorithm 3 outputs $\widehat{b}_i$ such that $1 - \mathcal{O}(\alpha))b_i \leq \widehat{b}_i \leq b_i$, where $b_i$ is the size of level set $i$.*

We then argue the privacy and space complexity of Algorithm 3.

**Lemma 4.8.** *Algorithm 3 is $\left(\frac{\varepsilon}{4}, \frac{\delta}{4}\right)$-differentially private for $\delta = \frac{1}{\text{poly}(m)}$ and uses space $\text{mmc}(L)^2 \cdot \text{poly}\left(\frac{1}{\alpha}, \frac{1}{\varepsilon}, \log m\right)$.*

## 4.4 PUTTING THINGS TOGETHER

We would like to combine the subroutines from the previous sections to output a private dataset for symmetric norm estimation. Thus it remains to describe how to privately partition the coordinates into the high, medium, and low frequency levels. To that end, we remark that although PRIVCOUNTSKETCH actually provides an estimated frequency for each coordinate, for our purposes, we only need estimated frequencies for the $L_2$-heavy hitters and there are at most $K := \mathcal{O}\left(\frac{1}{\eta^2}\right)$ possible $L_2$-heavy hitters with whichever threshold $\eta$ that we choose, e.g., $\eta = \alpha^2 \beta'$ in Algorithm 1. Thus it suffices to observe that we can privately partition the coordinates into the high, medium, and low frequency levels by first privately outputting the top $K$ estimated frequencies and then partitioning the coordinates according to their noisy estimated frequencies, which can be viewed as post-processing. In particular, (Qiao et al., 2021) observes that it suffices to add Laplacian noise with scale $\frac{8}{\eta \varepsilon}$ to each of the frequencies and then outputting the top $K$ noisy estimated frequencies to achieve $\frac{\varepsilon}{4}$-differential privacy. We now finally put together the results from the previous sections to show the following result. In particular, correctness follows from applying Lemma 2.7 to Lemma 4.1, Lemma 4.5, and Lemma 4.7, while privacy and the space complexity follow from Lemma 4.2, Lemma 4.6, and Lemma 4.8.

**Theorem 4.9.** *There exists a $(\varepsilon, \delta)$-differentially private algorithm that outputs a set $C$, for $\delta = \frac{1}{\text{poly}(m)}$. From $C$, the $(1 + \alpha)$-approximation to any norm with* maximum modulus of concentration *at most $M$ can be computed, with probability at least $1 - \delta$. The algorithm uses $M^2 \cdot \text{poly}\left(\frac{1}{\alpha}, \frac{1}{\varepsilon}, \log m\right)$ bits of space.*

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
