# OpenReview forum: "Private Data Stream Analysis for Universal Symmetric Norm Estimation"
_ICLR.cc/2023/Conference — Submitted to ICLR 2023_

### Official Review · Reviewer_9ZLt · 2022-10-23

**Confidence:** 2
**Correctness:** 3
**Technical Novelty And Significance:** 3
**Empirical Novelty And Significance:** Not applicable
**Recommendation:** 3

**Clarity, Quality, Novelty And Reproducibility:**

Clarity: The paper is somewhat clear, but some important details are missing or unclear.

Quality: The paper has minor technical flaws. For example, the experimental evaluation is weak.

Novelty: The paper contributes some new ideas or represents incremental advances.

**Details Of Ethics Concerns:**

One paper with the same title and abstract has been published at FORC2022 by authors <<REMOVED BY PCs>> I am not sure whether this violates the policy of ICLR.

**Strength And Weaknesses:**

Strength:
The targeted problem is interesting. This paper provides many theoretical results.

Weakness:
1. In general, the manuscript is not very well written. I noticed a lot of typographical and grammatical errors that made reading and understanding the manuscript a bit difficult. This paper is not easy to follow. There are many technical phrases that lack sufficient explanations.
2. No exact definition of (1+\alpha)-approximation in the main manuscript.
3. It would be better to provide PRICOUNTSKETCH instead of the sketching algorithms in Section 3.
4. The level thresholds T_1 and T_2 to separate the data is important. Algorithm 1 gives some theoretical order of the thresholds. However, how to choose them numerically?
5. There is no experimental evaluation of the proposed algorithms in this paper.

**Summary Of The Paper:**

This paper proposed a framework to release the family of symmetric norms with differential privacy. The mechanism permits (1+\alpha)-approximation to each norm.

**Summary Of The Review:**

This paper proposed a framework to release the family of symmetric norms with differential privacy.  However, some important details are missing or unclear.  I may have missed some crucial details.

---

> ### Comment · Program_Chairs · 2022-11-08
> **review edited to protect author identity**
>
> The above review is edited by PCs to protect author identity.

---

> > ### Comment · Program_Chairs · 2022-11-08
> > **-**
> >
> > PCs investigated FORC2022 submission that the reviewer raised above. We confirm that the venue is non-archival, which means it's not violation of ICLR policy. As such, ethical concerns raised in the above review is no longer a concern. Thank you.

---

> ### Author Response · Authors · 2022-11-10
> **Response to Reviewer 9ZLt**
>
> > No exact definition of (1+\alpha)-approximation in the main manuscript.
>
> Although we believe $(1+\alpha)$-approximation is standard terminology, we do agree that it should nonetheless be defined. Thanks for the feedback. We have added this definition to the full version of the paper.
>
> > It would be better to provide PRICOUNTSKETCH instead of the sketching algorithms in Section 3.
>
> The private version of CountSketch is a bit more involved since it also involves a private threshold subroutine. We want to use the private version of CountSketch as a black-box and thus we respectfully feel that a significantly expanded discussion in the extended abstract would detract from our algorithmic contributions. We do have a more detailed description of the private version of CountSketch in Section 2.2 of the full version in the supplementary material.
>
> > The level thresholds T_1 and T_2 to separate the data is important. Algorithm 1 gives some theoretical order of the thresholds. However, how to choose them numerically?
>
> There exist explicit values that the algorithm choose numerically. Specifically, note that Algorithm 1 sets $\beta'=\mathcal{O}\left(\frac{\alpha^2\beta\varepsilon^2}{\log^2 m}\right)$ or more specifically $\beta'=\frac{\alpha^2\beta\varepsilon^2}{2\log^2 m}$. Then $\beta'\cdot F_2(x)$ corresponds to the threshold $T_1$, which is utilized in the proof of Lemma 4.1. Similarly, Algorithm 2 leverages the quantity $\frac{\log n}{\beta'\alpha\varepsilon}$ to define the threshold $T_2$, which is then utilized in the proof of Lemma 4.4. We will expand the discussion surrounding $T_1$ and $T_2$ in the full version of the paper to include this additional intuition.
>
> > There is no experimental evaluation of the proposed algorithms in this paper...Quality: The paper has minor technical flaws. For example, the experimental evaluation is weak.
>
> Respectfully, we do not believe the absence of empirical evaluation equates to minor technical flaws. In particular, the goal of our paper was primarily to develop the first private, accurate, and efficient symmetric norm estimation algorithms with worst-case guarantees. We believe that empirical evaluation to confirm our theory would be valuable as future work but beyond the scope of our current paper.
>
> > One paper with the same title and abstract has been published at FORC2022 by authors <<REMOVED BY PCs>> I am not sure whether this violates the policy of ICLR.
>
> The ICLR program chairs have confirmed at https://openreview.net/forum?id=zGy_wqpRGTa&noteId=dDDbwTW8pCy that our submission does not violate the submission/ethics policies of ICLR.
>
> In summary, we believe we have answered your specific technical questions in your initial review and provided editorial adjustments to improve presentation. We also do not believe that the lack of experiments should mean that there are minor technical flaws with our paper. Finally, we emphasize that the ICLR program chairs have confirmed that there are no dual submission/ethics violation. Thus we believe we have addressed all of your initial concerns. In light of these points, we hope you will consider significantly raising your score. Otherwise, if you feel your concerns have not been sufficiently addressed or if you have any remaining concerns, could you please let us know so that we have a chance to respond? Thanks!

---

### Official Review · Reviewer_a8eC · 2022-10-24

**Confidence:** 3
**Clarity, Quality, Novelty And Reproducibility:** Yes
**Correctness:** 4
**Technical Novelty And Significance:** 4
**Empirical Novelty And Significance:** 4
**Recommendation:** 8

**Strength And Weaknesses:**

I found the idea of classifying and separately handling high, medium and low frequency levels novel and interesting. It makes sense to release approximate frequencies for the high frequency levels while release approximate level sizes for the light frequency levels for better utility. The theoretical analysis for the accuracy and privacy guarantee looks sound to me.

**Summary Of The Paper:**

The paper proposes a general algorithmic framework for differentially privately releasing a number of sufficient statistics on a data stream. With the released statistics, all symmetric norms can be approximated simultaneously, which makes the algorithm especially suitable for a large number of queries.

**Summary Of The Review:**


I found the idea of classifying and separately handling high, medium and low frequency levels novel and interesting. It makes sense to release approximate frequencies for the high frequency levels while release approximate level sizes for the light frequency levels for better utility. The theoretical analysis for the accuracy and privacy guarantee looks sound to me.

---

> ### Author Response · Authors · 2022-11-10
> **Response to Reviewer a8eC**
>
> Thank you for your feedback. We are greatly appreciative of your positive comments on our paper!

---

### Official Review · Reviewer_1krD · 2022-10-25

**Confidence:** 1
**Clarity, Quality, Novelty And Reproducibility:** See above.
**Correctness:** 4
**Technical Novelty And Significance:** 4
**Empirical Novelty And Significance:** 4
**Recommendation:** 6

**Strength And Weaknesses:**

This paper studies a very important problem, i.e., releasing the norms of data while protecting the differential privacy in streaming cases. Previous work many focus on limited settings (i.e., p<1), while this paper proposes a universal mechanism which works for any symmetric norms with low space complexity. This idea for coordinate parathion to my knowledge is novel and very interesting.

I have a hard time reading this paper. I think the following advices may be helpful:

1.	A more clear and detailed definition of the streaming model. Currently, it is not very clear to the readers what the exact streaming model is. The authors only mentioned the model with one sentence (“The simplest streaming model is perhaps the insertion-only model,…”). Is this the only model the authors study, or the model can be more general?

2.	The preliminary should be put before Section 2, since many definitions of Section 3 are used in Section 2.

3.	It would be great if the authors can add a related work section to compare with other work (especially smith et al., 2020, wang et al., 2021, blocki et al., 2012) and illustrate the differences.

4.	“We emphasize that prior to our work, there is no algorithm that can handle private symmetric norm estimation”I am not sure if it is an overclaim, since previous can handle the cases when p<1.


**Summary Of The Paper:**

This paper studies preserving differential privacy in when releasing information of streaming models. The authors propose a new DP mechanism that can release a set of sufficient statistics which can be used to estimate any symmetric norm of the data.

**Summary Of The Review:**

I think this paper is novel and the results are significant. But the writing can be improved.

---

> ### Author Response · Authors · 2022-11-10
> **Response to Reviewer 1krD**
>
> > A more clear and detailed definition of the streaming model. Currently, it is not very clear to the readers what the exact streaming model is. The authors only mentioned the model with one sentence (“The simplest streaming model is perhaps the insertion-only model,…”). Is this the only model the authors study, or the model can be more general?
>
> In particular, the model can be more general in the sense that our algorithm can handle both insertions and deletions of elements, i.e., the turnstile model. However, for the purposes of algorithmic intuition, it suffices to consider just the insertion-only model and indeed, our contributions are novel even for the insertion-only model.
>
> Thanks for the suggestion, we will expand the discussion of the various streaming models that we can handle in the full version of the paper.
>
> > The preliminary should be put before Section 2, since many definitions of Section 3 are used in Section 2.
>
> Yes, we agree -- this is a great suggestion. We have implemented this in the updated version, thanks!
>
> > It would be great if the authors can add a related work section to compare with other work (especially smith et al., 2020, wang et al., 2021, blocki et al., 2012) and illustrate the differences.
>
> Section 1.2 of the full version of the paper in the supplementary material focuses on related work and performs comparisons, both qualitatively and algorithmically, to previous works.
>
> > “We emphasize that prior to our work, there is no algorithm that can handle private symmetric norm estimation”I am not sure if it is an overclaim, since previous can handle the cases when p<1
>
> To be more clear, prior to our work, there is no algorithm that can handle general private symmetric norm estimation. Indeed there are several works that can handle private norm estimation of specific norms, but no work that handles a general private symmetric norm. For example, there is no work that handles $L_p$ estimation for $p>2$. Our work not only handles general private symmetric norm estimation, but it also does it simultaneously. That is, there is one private release of statistics from which an arbitrary number of symmetric norm estimation can simultaneously be performed.
>
> > I think this paper is novel and the results are significant. But the writing can be improved.
>
> Thanks for the feedback. We have implemented your suggestions for improving presentation and we believe our response addresses your initial comments. If you agree that your questions have all been answered satisfactorily, we hope that you will consider raising your score. Otherwise please let us know what remaining reservations you may have and we would be happy to engage in further discussion, thanks!

---

### Official Review · Reviewer_DEcx · 2022-11-01

**Confidence:** 3
**Correctness:** 3
**Technical Novelty And Significance:** 3
**Empirical Novelty And Significance:** Not applicable
**Recommendation:** 3

**Clarity, Quality, Novelty And Reproducibility:**

On page 1, please define more precisely “maximum modulus of concentration”.  The current definition refers to a median without saying from which set the median should be taken.

While in the beginning of Sec 1 x is called a vector, early in Sec 2 it is called a frequency vector (without clear definition for these frequencies).  The text next immediately starts discussing level sets without defining what level sets are being considered.

Section nicely defines classic notations such as [n] but then again is vague about “frequency vector x over a data stream”.  I suggest to give in one or two lines the structure of and notation for the data stream and the definition of x in terms of it (and possibly time parameters if the frequency only covers a certain time window).

Does the AMS algorithm choose \sigma uniformly at random?

In def 3.1, specify H\subseteq [n]
In def 3.1 (3), just setting \hat x_k = x_k seems to meet the requirement, nothing seems to say exact equality is not allowed.
In fact, def 3.1 seems to be more an algorithm than a definition.

Later, it becomes more and more difficult to follow the text as i get more and more uncertain about the definition of all concepts used.




**Details Of Ethics Concerns:**

—

**Strength And Weaknesses:**

Intuitively, the idea sounds very plausible to me, but the text is insufficiently rigorous for me to be able to follow the proof.

**Summary Of The Paper:**

This paper claims that sufficient statistics can be generated from which a wide range f norm estimations can be differentially privately computed.

**Summary Of The Review:**

The paper suggests a nice idea for which probably a proof exists, but the elaboration is insufficiently rigorous.

---

> ### Author Response · Authors · 2022-11-10
> **Response to Reviewer DEcx**
>
> > On page 1, please define more precisely “maximum modulus of concentration”. The current definition refers to a median without saying from which set the median should be taken.
>
> The maximum modulus of concentration is first mentioned in Theorem 1.2 on page 2, after which it is defined as the "median value of a norm on the $L_2$-unit sphere...". In other words, the median is over the $L_2$-unit sphere. We have also formally defined the concept later in Definition 3.5 and in even greater detail in Section 2.3 of the full version in the supplementary material.
>
> > While in the beginning of Sec 1 x is called a vector, early in Sec 2 it is called a frequency vector (without clear definition for these frequencies).
>
> At the end of the first paragraph, we state that in the streaming model, the vector $x\in\mathbb{R}^n$ is called a frequency vector when $x_i$ denotes the number of times $i$ has appeared in the stream.
>
> > The text next immediately starts discussing level sets without defining what level sets are being considered.
>
> We state at the bottom of the third page that the $i$-th level set consists of the coordinates with frequency in the range $[\xi^i,\xi^{i+1})$ for some $\xi>1$.
>
> > Section nicely defines classic notations such as [n] but then again is vague about “frequency vector x over a data stream”. I suggest to give in one or two lines the structure of and notation for the data stream and the definition of x in terms of it (and possibly time parameters if the frequency only covers a certain time window).
>
> As previously stated, we describe how the frequency vector $x$ is defined by the streaming model at the end of the first paragraph.
>
> > Does the AMS algorithm choose \sigma uniformly at random?
>
> For the AMS algorithm, it suffices to choose $\sigma_i\in\{-1,+1\}$ to be pairwise independent. However, we remark that not only is this approach standard, but also that the AMS algorithm is used as a black-box for our algorithm.
>
> > In def 3.1, specify H\subseteq [n] In def 3.1 (3), just setting \hat x_k = x_k seems to meet the requirement, nothing seems to say exact equality is not allowed. In fact, def 3.1 seems to be more an algorithm than a definition.
>
> Definition 3.1 is not an algorithm; it is the definition of the heavy-hitters problem. The goal is to output an estimate $\widehat{x_k}$ of the true value $x_k$. Of course outputting $x_k$ would suffice but the challenge for heavy-hitters in the streaming model is that space limitations often do not permit $x_k$ to be explicitly maintained.
>
> > Intuitively, the idea sounds very plausible to me, but the text is insufficiently rigorous for me to be able to follow the proof...Later, it becomes more and more difficult to follow the text as i get more and more uncertain about the definition of all concepts used. The paper suggests a nice idea for which probably a proof exists, but the elaboration is insufficiently rigorous.
>
> It currently seems that all concerns raised in the review have already been addressed within the text of the initial submission. Thus without additional concerns about specific details, we hope that you will strongly consider significantly increasing your score, as we do not believe your current score of 3 is representative, especially given that the review (1) seems to majorly overlook the conceptual contribution of our results and (2) focuses largely on claimed presentation shortcomings that do not align with the details already in the initial submission. Otherwise, if there are additional questions or comments, we would be happy to clarify any points of confusion since we do believe our work is quite technically involved. Thanks!

---

> > ### Comment · Reviewer_DEcx · 2022-11-13
> > **Response to author response**
> >
> >
> >
> > >>  On page 1, please define more precisely “maximum modulus of concentration”. The current definition refers to a median without saying from which set the median should be taken.
> >
> > > The maximum modulus of concentration is first mentioned in Theorem 1.2 on page 2, after which it is defined as the "median value of a norm on the -unit sphere...". In other words, the median is over the-unit sphere. We have also formally defined the concept later in Definition 3.5 and in even greater detail in Section 2.3 of the full version in the supplementary material.
> >
> > After definition I don't read "median value of a norm on the -unit sphere" but "measures the worst-case ratio of the maximum value to the median value of a norm on the L2 -unit sphere for any restriction of the coordinates".
> > We could read it as "the worst-case ratio of (the maximum value over the median value of a norm) on the L2-unit sphere"  So we are taking the "worst case ratio" (this may be the highest or the lowest value depending on your point of view, the text doesn't specify) over the L2 sphere of a value which is equal to "the maximum value to the medial value of a norm".  So, we need to take worst-case_{x \in L2-unit sphere} (max_y f(x,y) / median_y n(x,y)).  However, it is unclear over what set y ranges to take the maximum and median.  Since we have TWO aggregation operator ("worst case" and "median") and only one specification of a set over which to aggregate, at least one of both aggregation operators has no set specified.
> > Maybe the authors parse this sentence differently, in which case the grammar is ambiguous.
> >
> > After definition 1.2, the text does not point the reader to the full version of the text for more details.
> >
> > >>  While in the beginning of Sec 1 x is called a vector, early in Sec 2 it is called a frequency vector (without clear definition for these frequencies).
> >
> > > At the end of the first paragraph, we state that in the streaming model, the vector x is called a frequency vector when x_i denotes the number of times i has appeared in the stream.
> >
> > Unfortunately, I don't find this definition at the first use of "frequency vector".  In particular, "frequency vector" is used once in Section 1 in "is perhaps the insertion-only model, in which a sequence of m updates increments coordinates of an n-dimensional frequency vector x and the goal is ...", but here is no such definition.  The word "frequency" next occurs in Section 2 in the second paragraph, in "For a (1 + α)-approximation,  the algorithm partitions the n coordinates of the frequency vector x into powers of ξ-based on their magnitudes ...".  Let's ignore that "ξ-based" isn't defined nor grammatically correct.   Again, no definition for "frequency vector" is provided here.  The word "frequency" next occurs in the third paragraph of Section 2, but not in the phrase "frequency vector".  Still, here too no definition of "frequency" is provided.
> >
> > Even if indeed the explanation proposed by that authors ", the vector x is called a frequency vector when x_i denotes the number of times i has appeared in the stream." is present in the text, it would be more rigorous to specify (a) that i is an item (and the set of all items in {1,...n}), (b) the stream can be represented as a set/vector/..., (c) how frequency is defined (please note the field of frequent pattern mining has quite a few papers defining frequency in different ways), etc.
> >
> > >>  The text next immediately starts discussing level sets without defining what level sets are being considered.
> >
> > > We state at the bottom of the third page that the i-th level set consists of the coordinates with frequency in the range for some
> >
> > I see at least 7 occurrences of "level set" of page 3, so if you define it first at the bottom of page 3 it is not defined at the first occurrence of the concept.
> > Moreover, the sentence you refer to reads "However, the level sets consisting of elements with frequencies between [ξ^ i , ξ^{i+1} ) for small i ...".
> > Please notice that this sentence does NOT contain "," before and after the relative clause so the relative clause is non-defining (=restricting) here.  It says that the sentence only applies to the level sets which happen to consist of elements with frequencies between ...
> > Moreover, please notice that before "elements" we don't have a quantifier.  IT doesn't say "all elements" or "some elements" or just "2 elements", we only know that the sentence applies to level sets containing such elements (it is plural but in mathematics we often use plural when we mean "0,1,2 or more" too).
> >
> > In conclusion, this is not a definition, and it is not presented at the first use of the concept.  The reader first should wonder what "level set" means in your paper until after the 7th use of the term he gets an incomplete / non-rigorous description.

---

> > ### Comment · Reviewer_DEcx · 2022-11-13
> > **response to author response (2)**
> >
> >
> > >>    Section nicely defines classic notations such as [n] but then again is vague about “frequency vector x over a data stream”. I suggest to give in one or two lines the structure of and notation for the data stream and the definition of x in terms of it (and possibly time parameters if the frequency only covers a certain time window).
> >
> > > As previously stated, we describe how the frequency vector is defined by the streaming model at the end of the first paragraph.
> >
> > As previous stated, you are too optimistic.
> >
> > >>  Does the AMS algorithm choose \sigma uniformly at random?
> >
> > > For the AMS algorithm, it suffices to choose sigma_i to be pairwise independent. However, we remark that not only is this approach standard, but also that the AMS algorithm is used as a black-box for our algorithm.
> >
> > It is better to make the paper self-contained.  It seems helpful for the reader to know the probability distribution.
> >
> > >>  In def 3.1, specify H\subseteq [n] In def 3.1 (3), just setting \hat x_k = x_k seems to meet the requirement, nothing seems to say exact equality is not allowed. In fact, def 3.1 seems to be more an algorithm than a definition.
> >
> > > Definition 3.1 is not an algorithm; it is the definition of the heavy-hitters problem. The goal is to output an estimate
> > of the true value . Of course outputting would suffice but the challenge for heavy-hitters in the streaming model is that space limitations often do not permit
> >
> > A definition is normally introducing a mathematical concept.  Definition 3.1 contains the word "compute" a hence seems to also give instructions to the reader or a computer.
> >
> > The reply doesn't answer my main comment that "In def 3.1 (3), just setting \hat x_k = x_k seems to meet the requirement, nothing seems to say exact equality is not allowed. I".  In particular (3) says we should compute and approximation,    This makes the meaning of the definition unclear.  Rereading the definition it seems that probably one or more grammatical mistakes may have cause the text to have a different meaning than what you intend.
> >
> > Please note that the term "heavy hitter" is used several times before definition 3.1 without a definition of its own.  Informally we can understand what is a heavy hitter, but if you want to present a formal definition of a property of a heavy hitter it would seem natural to first present a formal definition of heavy hitter itself.    If space is too limited, you may put less essential details in the supplementary material, but in that case please do refer to that supplementary material in a clear way.  Please consider that the call for paper explicitly says "Authors may use as many pages of appendices (after the bibliography) as they wish, but reviewers are not required to read the appendix." where both parts of the sentence are important.
> >
> > > It currently seems that all concerns raised in the review have already been addressed within the text of the initial submission. Thus without additional concerns about specific details, we hope that you will strongly consider significantly increasing your score, as we do not believe your current score of 3 is representative, especially given that the review (1) seems to majorly overlook the conceptual contribution of our results and (2) focuses largely on claimed presentation shortcomings that do not align with the details already in the initial submission. Otherwise, if there are additional questions or comments, we would be happy to clarify any points of confusion since we do believe our work is quite technically involved. Thanks!
> >
> > While the authors believe all concerns were already addressed in the original submission, in fact none of them were already addressed.  I can only conclude that the authors are too optimistic, we can't trust they will improve the presentation of the paper by the final version and hence rejection is the safest option.
> >
> > I unfortunately don't have sufficient time to list each and every presentation issue or grammatical mistake leading to confusion about the meaning of sentences, especially if a lot of discussion is needed before it is clear why sentences are unclear or definitions are missing.  I can only recommend to carefully proofread the text and imagine how a reader who does not have any prior knowledge may experience it.

---

### Author Response · Authors · 2022-11-16
**Update**

Hi everyone,

**We have uploaded new versions of the extended abstract and the full version of the paper in the supplementary material**. For the sake of clarity, we have highlighted all changes in the full version in blue.

For the extended abstract, we reordered the preliminary and technical overview sections, per the suggestion of Reviewer 1krD. For the full version in the supplementary material, we have incorporated the feedback of all reviewers, including additional details on the streaming model and induced frequency vector, the value of thresholds $T_1$ and $T_2$, the definition of approximation algorithms, level sets, and the AMS algorithm.

As the discussion period is drawing to a close, we wanted to check whether the reviewers (or the area chair) had any remaining unresolved questions or concerns that we could potentially address.

Thanks again for your consideration!

---

### Decision · Program_Chairs · 2023-01-20

**Decision:**

Reject

**Justification For Why Not Higher Score:**

Writing is very hard to follow and might require re-writes. Lack of empirical evaluation.

**Justification For Why Not Lower Score:**

N/A

**Metareview: Summary, Strengths And Weaknesses:**

Paper proposes a novel mechanism to release symmetric norms in a differential private manner. Authors propose an interesting idea to separately handle high "frequency"  and low "frequency" dimensions. Former dimensions are approximately revealed while latter dimensions are revealed through level sizes. This way the proposed algorithm is able to release sufficient statistics for all Lp norms at one go. Unfortunately the paper is very hard to read. For example, notations are used before defining (e.g. mmc) and some of the vaguely defined terms may not be standard or familiar to ICLR community (e.g. "frequency vector"). In the response to reviewers authors claim that
> we state that in the streaming model, the vector x is called a frequency vector when x_i denotes the number of times i has appeared in the stream.

However, this statement wasn't found in the main text. Discussion of related work is absent from main text. Paper might need a re-write to relegate unimportant lemmas/results to appendix to make space for discussion for clarity. I also highly encourage the authors make the main text and supplementary text of the same format in terms of section ordering, numbering and citation style. Current mismatch is very hard to follow. It is also noted the paper does not provide any empirical results and validations.